# Validation of a Genotype-Independent Hepatitis C Virus Near-Whole Genome Sequencing Assay

**DOI:** 10.3390/v13091721

**Published:** 2021-08-30

**Authors:** Hope R. Lapointe, Weiyan Dong, Winnie W. Y. Dong, Don Kirkby, Conan Woods, Art F. Y. Poon, Anita Y. M. Howe, P. Richard Harrigan, Chanson J. Brumme

**Affiliations:** 1Department of Medicine, Division of Social Medicine, University of British Columbia, Vancouver, BC V6T 1Z4, Canada; hope.lapointe@ubc.ca (H.R.L.); richard.harrigan@ubc.ca (P.R.H.); 2BC Centre for Excellence in HIV/AIDS, Vancouver, BC V6Z 1Y6, Canada; edong@bccfe.ca (W.D.); wdong@bccfe.ca (W.W.Y.D.); dkirkby@bccfe.ca (D.K.); email@woodsc.ca (C.W.); 3Department of Pathology and Laboratory Medicine, Western University, London, ON N6A 3K7, Canada; apoon42@uwo.ca; 4British Columbia Centre for Disease Control, Vancouver, BC V5Z 4R4, Canada; anitahowe@live.com; 5Department of Medicine, Division of Infectious Diseases, University of British Columbia, Vancouver, BC V6T 1Z4, Canada

**Keywords:** HCV, direct-acting antiviral agent, resistance-associated substitutions, whole-genome sequencing, genotype-independent

## Abstract

Despite the effectiveness of direct-acting antiviral agents in treating hepatitis C virus (HCV), cases of treatment failure have been associated with the emergence of resistance-associated substitutions. To better guide clinical decision-making, we developed and validated a near-whole-genome HCV genotype-independent next-generation sequencing strategy. HCV genotype 1–6 samples from direct-acting antiviral agent treatment-naïve and -treated HCV-infected individuals were included. Viral RNA was extracted using a NucliSens easyMAG and amplified using nested reverse transcription-polymerase chain reaction. Libraries were prepared using Nextera XT and sequenced on the Illumina MiSeq sequencing platform. Data were processed by an in-house pipeline (MiCall). Nucleotide consensus sequences were aligned to reference strain sequences for resistance-associated substitution identification and compared to NS3, NS5a, and NS5b sequence data obtained from a validated in-house assay optimized for HCV genotype 1. Sequencing success rates (defined as achieving >100-fold read coverage) approaching 90% were observed for most genotypes in samples with a viral load >5 log_10_ IU/mL. This genotype-independent sequencing method resulted in >99.8% nucleotide concordance with the genotype 1-optimized method, and 100% agreement in genotype assignment with paired line probe assay-based genotypes. The assay demonstrated high intra-run repeatability and inter-run reproducibility at detecting substitutions above 2% prevalence. This study highlights the performance of a freely available laboratory and bioinformatic approach for reliable HCV genotyping and resistance-associated substitution detection regardless of genotype.

## 1. Introduction

Hepatitis C virus (HCV) infection is a major public health concern in Canada and worldwide. In 2015, the World Health Organization (WHO) estimated that 71 million people globally had chronic HCV infection [1]. Nearly 400,000 die annually from HCV infection, largely due to liver cirrhosis and hepatocellular carcinoma. Despite an estimated 843,000 individuals being cured in 2015, these estimates are in sharp contrast to the number of new HCV infections globally (1.75 million). Without appropriate treatment scale-up and therapeutic monitoring, the global HCV epidemic may continue to expand in severity.

The therapeutic approach towards HCV chronic infection has shifted in recent years with the advent of oral direct-acting antiviral (DAA) agents. Previously, first-line treatment options were limited to pegylated interferon-, ribavirin- and first-generation protease inhibitor-containing regimens [2]. Interferon- and ribavirin-based therapies were non-specific to HCV and caused moderate to severe side effects universally, while first-generation protease inhibitors also had moderate to high potential for drug–drug interactions [3]. Newer therapies against specific viral targets use direct-acting antiviral agents (DAA), small-molecule inhibitors that can be categorized on the basis of their viral targets. These drug targets are important for viral replication, notably non-structural proteins NS3/4a, NS5a, and NS5b.

Some DAA-based regimens are genotype-specific, requiring personalized therapy to ensure genotype coverage by antiviral agents. The recent introduction of pan-genotypic options (sofosbuvir/velpatasvir, glecaprevir/pibrentasvir) circumvent the need for genotype identification [4,5]. These pan-genotypic regimens are now recommended by the WHO as first-line therapy for all adults with HCV infection regardless of genotype [6,7]. With this highly effective repertoire of DAA medications, overall sustained virologic response rates have demonstrated >90–95% regardless of HCV genotype, baseline HCV RNA viral loads, race, HIV coinfection and hepatic fibrosis [8,9,10,11].

However, cases of DAA treatment failure have been associated with the emergence of drug-resistance-associated substitutions (RAS) in all HCV genotypes, detected in both DAA treatment-naïve and treatment-experienced people [12,13,14,15]. The failure to achieve sustained virologic response through the emergence of RAS is thought to be the result of the selective pressure of antiviral drugs during treatment or the genetic variation inherent in the virus itself. As such, the identification of RAS can enable clinicians to personalize DAA treatment for a given patient, typically through the addition of ribavirin or by extending therapy duration, thereby enhancing the likelihood of treatment success.

Currently, the use of drug resistance testing in clinical management of HCV differs slightly between national and international societies. The Canadian Association for the Study of the Liver (CASL), the European Association for the Study of the Liver (EASL) and the American Association for the Study of Liver Diseases (AASLD) recommend resistance testing in select clinical scenarios [16,17,18]. As most HCV-positive individuals achieve sustained virologic response, baseline resistance testing is not universally recommended for all HCV infections [16,17]. RAS identification may be recommended in DAA-treatment-naïve individuals, though almost solely for genotype 1a and 3 infections, dependent on treatment status (naïve vs. experienced), the selected treatment regimen and cirrhosis status. Resistance testing is generally not recommended in patients with other HCV genotypes (1b, 2, 4, 5 and 6). Drug resistance testing can be considered in cases of treatment failure and retreatment, though limited data are available on the clinical utility of this approach. HCV resistance testing nevertheless provides an opportunity to identify RAS, notably in the case of genotype 1a or 3 infections as well as previously treated patients who have failed an initial DAA regimen. Such testing can also be used in the context of epidemiological surveillance of HCV drug resistance to provide valuable insight into treatment and prevention strategies [19]. Drug resistance testing therefore may continue to provide decisional support for certain clinical scenarios, such as optimizing second-line therapy [20].

To our knowledge, there is no commercially available kit to perform HCV drug resistance testing for all marketed DAAs. For testing, clinicians and/or public health labs must submit samples to one of the few select labs with in-house developed assays for clinical HCV RAS identification (e.g., BC Centre for Excellence in HIV/AIDS, Quest Diagnostics, Labcorp). For instance, HCV clinical specimens in Canada are first submitted to a provincial public health laboratory, which subsequently refers the sample to the BC Centre for Excellence for HIV/AIDS for HCV drug resistance testing. Currently, there are a small number of whole-genome sequencing approaches published for HCV, though these have widely variable performance and uses, may not have been fully validated with clinical isolates and may be technically incompatible with certain HCV genotypes or impractical for clinical use [21,22,23,24,25,26,27,28,29,30]. Bioinformatic challenges also represent a notable hurdle in this field, which currently lacks a standardized approach to analyzing next-generation sequencing data. Few open-source software for genome sequencing and drug resistance testing are available, with alternatives typically presenting with technical and financial barriers to their use.

We therefore developed and validated a near-whole genome, HCV genotype-independent sequencing assay using the MiSeq platform (Illumina) to identify RAS that may be associated with treatment failure. This assay also allows for HCV genotype classification using genomic data in lieu of conventional probe-based assays. The results produced by this assay can help guide clinicians and patients when making treatment decisions in specific clinical scenarios, as outlined by treatment guidelines. Other public health laboratories globally can adopt this HCV drug resistance assay, for which an associated bioinformatic pipeline is available specifically for generating HCV drug resistance reports.

## 2. Materials and Methods

### 2.1. Study Population

Sample sets for each assay performance parameter are outlined in Table 1. Briefly, HCV-positive samples used to develop and validate this near whole genome, HCV genotype-independent assay were sourced from Merck from the C-WORTHY trial (NCT01717326/Protocol PN035) and the Vancouver Injection Drug Users Study (VIDUS) [31,32]. Samples included specimens from treatment-naïve HCV-infected individuals, as well as DAA-treated persons who did not achieve sustained virologic response at 12 weeks post-treatment. All HCV-positive samples were tested for genotype (GT) 1–6 by sample providers themselves using the VERSANT HCV Genotype 2.0 line probe assay (LiPA; Siemens, Erlangen, Germany).

### 2.2. PCR Amplification

Viral RNA was extracted from 500 µL of plasma on a NucliSENS easyMag RNA/DNA Extractor (bioMérieux, Montréal, QC, Canada) according to manufacturer’s protocol. Extracted nucleic acids were eluted in 60 µL of elution buffer and stored at −20 °C until reverse-transcription-polymerase chain reaction (RT-PCR). A negative control (DEPC-treated water) was included in each run.

Nucleic acids underwent RT-PCR using SuperScript III Reverse Transcriptase (18080-044, Invitrogen, Carlsbad, CA, USA), followed by a nested PCR using Klentaq LA (110, DNA Polymerase Technology Inc., St. Louis, MO, USA) to amplify the HCV genome in two fragments (Figure 1).

Firstly, one near-whole-genome amplicon (“WG amplicon”) was generated spanning the HCV Core to NS5b codon 336 (GT1a_H77 genome coordinates 342 to 8610). To cover the remaining NS5b portion, a smaller, partially-overlapping amplicon (“MiDi amplicon”) was designed spanning NS5b codons 228 to the 3′ end of NS5b (H77 genome coordinates 7829 to 9377). Nested PCR products were visualized by agarose gel electrophoresis to confirm successful amplification prior to library generation. Amplicons were normalized and purified using AMPure XP magnetic beads (A63880, Beckman Coulter, Mississauga, ON, Canada). Purified amplicon concentrations were quantified using the Invitrogen Quant-iT Picogreen dsDNA assay (P7589, Invitrogen, Carlsbad, CA, USA) and diluted to 1 ng/μL. Note that the amplicons were processed separately until the pooling of the amplicon libraries in preparation for sequencing. Primers for reverse transcription and amplification steps are included in the Appendix A (Laboratory Protocol, Appendix A).

### 2.3. Library Preparation

Individual libraries for each amplicon (MiDi and WG amplicons) were prepared using the Nextera XT DNA Library Preparation Kit (FC-131-1024, Illumina) and Nextera XT Index Kits (FC-131-1002, Illumina) for amplicon tagmentation and dual-index barcoding, respectively. Libraries were prepared according to manufacturer protocols, with a modified tagmentation step. Here, 40% working volumes were used for the tagmentation reagents with 2 ng of DNA as input (see Appendix A–Methods). Indexed amplicons for each target were pooled and purified using AMPure XP beads, and the pooled library was diluted to 1.3 ng/µL. The WG and MiDi pooled libraries were then combined at a roughly 7:1 ratio, based on the approximate amplicon length. Note that the WG and MiDi amplicons from the same samples are not tagged with the same indices; unique index tag combinations are used for each amplicon, allowing for the assessment of sequence concordance in the overlapping regions. The Illumina PhiX control library was then added at 10% ratio to the sample DNA libraries and sequenced on the Illumina MiSeq platform using the 2 × 250 bp MiSeq Reagent Kit v2 (MS-102-2003, Illumina, San Diego, CA, USA). A complete standard operating procedure outlining the laboratory protocol for this assay is available in the Appendix A.

### 2.4. Data Processing

Raw short-read MiSeq data were processed by an in-house pipeline designed for rapidly evolving RNA viruses (MiCall v6.8, available at https://github.com/cfe-lab/MiCall and on Illumina BaseSpace Sequence Hub). Briefly, the MiCall pipeline uses bowtie2 to construct consensus sequences and estimate nucleotide frequencies following several steps: (1) initial read mapping to a set of HCV reference genome “seed” sequences, (2) sample-specific reference construction, remapping and reconstruction, (3) an iterative mapping process using sample-specific references, and (4) consensus sequence generation [33]. Given the extreme intra- and inter-genotype sequence diversity displayed by HCV, multiple reference sequences are used in the pipeline to maximize the number of individual MiSeq reads that are successfully mapped to a reference.

Firstly, paired-end reads are mapped to 57 HCV genotype/subtype reference sequences curated by the Los Alamos National Laboratory HCV Sequence Database [34]. After this preliminary mapping stage, HCV genotype references having ≥10 mapped reads are retained. Multiple different reference genotypes (e.g., GT1a, GT1b, GT2b) may be selected during this preliminary mapping; however, if more than one “seed” reference from the same genotype (e.g., GT1a, GT1b) is retained, only the seed with the most mapped reads is retained for that genotype.

Secondly, for each retained reference sequence, mapped reads are collapsed into a sample-specific reference that retains the genotype identity of the original “seed” Los Alamos reference. All reads are then remapped to each constructed sample-specific reference, which are then reconstructed using newly mapped reads.

Thirdly, reference-building and re-mapping described in the last step occurs iteratively until ≥95% of reads map to the sample-specific reference, or no additional reads are mapped in additional rounds of re-mapping. As this process continues and additional reads are mapped, it is expected that individual sample-specific references may diverge away from their initial “seed” reference sequence. For example, it is possible for a GT2b sample-specific reference to sufficiently change to more closely resemble a GT1a reference as re-mapping occurs, and new reads are iteratively incorporated. To address this, each iteration includes a genotype check for each sample-specific reference. Briefly, a genetic distance metric (Levenshtein distance) is computed between sample-specific references and all Los Alamos reference sequences. In instances where sample-specific references have drifted to more closely resemble another genotype than the “seed” genotype, the sample-specific reference is excluded. For instance, the GT2b sample-specific reference could be excluded here, and a more suitable GT1a sample-specific reference is retained for reference-building and re-mapping. In instances where the sample-specific references do not drift away from the expected genotype, they are all retained through to the end of the pipeline. Should this occur, the pipeline outputs multiple consensus sequences from a single sample, one per sample-specific reference. Paired-end reads are merged, and error correction rules are applied; corrections included discarding nucleotides with a quality score <15 and resolving mismatching base calls in overlapping reads by retaining the higher quality base.

Lastly, all paired-end reads are used to generate nucleotide consensus sequences at various nucleotide prevalence thresholds (1%, 2%, 5%, 10%, 20%, 25%); nucleotide mixtures were called when two or more bases were present above the prevalence cutoff. Amino acid frequencies and sequences were translated directly from paired-end reads and used for RAS analysis. Finally, the MiCall pipeline merges sequence data from the two amplicons WG and MiDi to generate single, near-full genome sequences. The software resolves the overlapping region between the WG and MiDi amplicons by retaining only the data from the amplicon with the greatest coverage. The genotype of the final assembled consensus is inferred again by calculating the shortest Levenshtein distance to the Los Alamos reference set. The resulting dataset covers the near whole-genome of HCV. To demonstrate that both amplicons generate concordant sequence data, we examined nucleotide agreement and RAS quantification within the overlapping segment of both products.

### 2.5. Quality Control Criteria

MiCall incorporates a final quality control step that verifies sequence coverage check at key resistance-associated positions. Samples with >1000-fold coverage at all RAS positions are automatically approved. Sequences with lower read coverage (100 to 999 reads) at ≥1 RAS position are flagged for manual review. Any sequences with fewer than 100 reads at any RAS position are considered to have failed and are excluded from downstream data analyses. Additionally, at least one of the sample’s gene regions (e.g., NS3, NS5a, or NS5b) must have at least 100-fold coverage at all RAS positions or will be deemed a failure.

### 2.6. Genotype Inference & RAS

For analytical purposes, sample genotypes were inferred from one of two sources, MiSeq genomic sequence data and LiPA assay results. Where sequencing was successful, genotypes were inferred according to the data processing outlined above. In instances where no MiSeq sequence was generated during this validation and HCV genotypes could not be inferred, viral classification was based on the LiPA assay genotype performed by the sample provider. Note that all genotypes listed in Table 1 are LiPA-based results.

Based on the inferred genotype, nucleotide consensus sequences were aligned to appropriate United States Food and Drug Administration (FDA) reference strain sequences for downstream identification of RAS and indels (Appendix A). RAS positions were restricted to positions and substitutions listed in Appendix A, adapted from known resistance-associated mutations at the time of analysis [35].

### 2.7. Assessment of Assay Performance

Performance metrics similar to our previous validation were used to assess assay quality between replicates and/or laboratory methods: (1) the concordance of nucleotide base calls, (2) the concordance of amino acid residue calls, (3) the quantification of RAS prevalence [36] and (4) sequencing success.

Concordance is defined as the proportion of nucleotide or amino acid sequence agreement between all nucleotides or amino acids sequenced across all samples. In cases of compatible discordances, (i.e., where one sequence observed a nucleotide or amino acid mixture whereas the other sequence observed a single component thereof, e.g., nucleotides Y vs. C), they were weighed identically as complete discordances. These concordance metrics were assessed using consensus sequences generated from specific nucleotide prevalence cutoffs.

Sequencing success was calculated as the proportion of sequences having met MiCall quality control criteria outlined above, requiring a minimum 100-fold coverage at all RAS-associated positions in NS3, NS5a and NS5b.

Unless otherwise stated, analyses were performed individually for NS3, NS5a and NS5b gene segments and used a 20% nucleotide or amino acid prevalence threshold for mixture calling, where applicable. A 20% cutoff was selected to mimic Sanger sequencing detection thresholds and is well above the expected minimum detected level [37]. In analyses where sequence data were generated from multiple sample replicates, and no gold-standard sample reference sequence was available, a consensus sequence was constructed from all available replicates for use as a comparator. Nucleotides and amino acids appearing in ≥20% of replicate sequences were included as mixtures in the consensus. In addition to the metrics described above, we also assessed concordance in the detection of all nucleotide substitutions across the genome at their respective prevalence.

#### 2.7.1. Accuracy

Accuracy, the level of agreement between a new test and a reference test, was assessed by comparing newly generated sequences to data obtained from a previously validated in-house assay optimized for HCV GT1 on the same Illumina MiSeq platform [36]. Duplicates of 93 samples were used here, consisting mostly of GT1 samples.

#### 2.7.2. Precision

Precision refers to the ability of an assay to reproduce its result, whether within or between tests. Repeatability (intra-run variability) was assessed using 12 replicates of four GT1 HCV-positive samples, processed together through amplification and sequenced in a single MiSeq run. Reproducibility (inter-assay variability) was assessed using five replicates of 12 samples (*n* = 11 GT1, *n* = 1 GT3), with each set of replicates processed separately in five PCR batches and sequenced on five separate days. To assess sequencing precision, sequences obtained from individual replicates were compared for concordance against a consensus sequence constructed from all available replicate sequences.

#### 2.7.3. Sensitivity

Sensitivity refers to the ability to reliably detect an analyte with acceptable precision. We assessed the HCV viral load limit of detection with two sample sets: (1) the same GT1 samples used for accuracy measurement (*n* = 93) and (2) a set of largely GT3-6 samples (*n* = 55).

The ability to reliably detect low prevalence substitutions or minority variants was assessed using two replicates of 95 clinical samples of multiple genotypes.

#### 2.7.4. Specificity

Analytical specificity refers to the ability of an assay to detect only the intended target and the lack of interference from other analytes. Specificity was assessed in two ways: (1) HCV-negative samples were assayed separately in triplicate and (2) a negative control (DEPC-treated water) was included in all runs for this validation and during routine clinical runs for months following this validation. We did not study interfering substances (e.g., hemoglobin, bilirubin, triglycerides, etc.) since our method of RNA extraction has been shown not to be affected by these substances [38]. In addition, we did not assess substances that may otherwise inhibit or interfere with RT-PCR amplification as these were previously assessed during the validation of an HCV whole-genome NGS method designed for GT1 infections [36,39]. It is reasonable to assume that the prior results would apply to the genotype-independent method as well given the minimal differences between the assays.

#### 2.7.5. Genotype Spectrum

The genotype spectrum of our assay was evaluated using the same GT1 (*n* = 93) and GT3-6 samples (*n* = 55) described above for accuracy and sensitivity. Laboratory staff were blinded to the expected, previously determined probe-based genotype for 37 (67%) of the GT3-6 samples.

## 3. Results

### 3.1. Accuracy

A total of 93 samples (predominantly GT1a) were processed in duplicate by the genotype-independent and GT1-optimized methods and used to assess accuracy, using the latter as the reference comparator. The amplification and sequencing of all three genes (NS3, NS5a and NS5b) in at least one replicate was successful for 88 (94.6%) and 85 (91.4%) samples by the genotype-independent and GT-1-optimized methods, respectively. For method comparisons, 85 samples (89.2%) had available data at all three genes by at least one replicate for both assays. In instances wherein a method generated sequence data by both replicates, only the first replicate’s data was analyzed for that respective method.

As the GT1-optimized assay was designed to amplify and sequence GT1a and GT1b specimens, sequence comparisons between methods were restricted to the successfully sequenced samples of the same genotype, leaving 83 samples for comparison of all three genes. The genotype-independent HCV method showed 99.85%, 99.86% and 99.90% nucleotide concordance with the HCV GT1-optimized NGS method in NS3, NS5a and NS5b, respectively, using a 20% mixed base-calling threshold. Sequence discordances represented 0.14% of the total bases compared (*n* = 489/352,418) and were entirely due to nucleotide mixtures identified by one method and not the other; no incompatible nucleotide discordances were observed. Importantly, neither method was systematically biased towards identifying minority variants. While the genotype-independent method identified a marginally larger proportion of the discrepant mixtures (*n* = 288; 58.9%), overall agreement in mixture calling was very high (Cohen’s κ = 0.94). Nucleotide concordance matrices between the two NGS methods, as well as between replicates of the genotype-independent method are presented in the Appendix A. Amino acid sequences were nearly identical for each gene, achieving ≥99.92% concordance.

RAS quantification was assessed for samples with >1000-fold coverage for both replicates in both assays. Samples with no detectable RAS at >2% prevalence by either NGS method were excluded. Overall we observed a strong linear correlation between the RAS population prevalence observed by both methods (R^2^ > 0.99) for NS3 and NS5a (Figure 2A). A systematic bias in amplifying specific RAS was not observed (Figure 2B). Linear regression could not be performed reliably for NS5b, but amino acid substitutions at positions 159, 282 and 320 were highly conserved (near 100% prevalence for all residues).

### 3.2. Precision

Both the reproducibility and repeatability of all possible amino acid substitutions across NS3, NS5b and NS5a were evaluated in the same manner. Precision metrics assessed the genotype-independent assay’s ability to detect nucleotide and amino acid substitutions consistently and quantitatively between replicates of a sample. For sequence concordance analyses, individual replicate sequences were compared to a sample consensus sequence constructed from the replicate sequences, using a 20% cutoff for mixture-calling.

We also assessed amino acid substitution detection at various prevalence categories. For each amino acid substitution, the mean prevalence across sequencing replicates was calculated and binned into <1%, 1–2%, 2–5%, 5–10%, 10–20% and >20% groups. The variant detection rate was then defined as the percent of sequenced replicates in which the amino acid was detected with a frequency above one of two minimum frequency thresholds: a 2% prevalence threshold was used to assess the potential for NGS sequencing to detect minority variants, while a 20% threshold was used to mimic Sanger sequencing detection limits.

### 3.3. Repeatability

Repeatability (intra-run variability) was assessed using 12 replicates of four GT1 HCV-positive samples. Nearly all replicates (42/48, 87.5%) were successfully sequenced by the genotype-independent method at all three genes, NS3, NS5a, NS5b. Note that five of the observed replicate sequencing failures were in NS5b by a single sample. Nucleotide and amino acid concordance relative to the consensus sequences was high for all samples; median nucleotide and amino acid concordance was above ≥99.90% and ≥99.93% respectively, across NS3, NS5a and NS5b.

At a 2% cut-off for mixture calling, the amino acid substitutions with a mean prevalence >5% were detected in 100% of the replicates in NS3, NS5A and NS5B. (Figure 3A). For substitutions with a mean prevalence between 2% and 5%, the repeatability of detection for NS3 and NS5A was ~80%; lower repeatability (~50%) was observed in NS5B. As expected, there was a significant drop in the detection rates at mean prevalence around or below the 2% variant threshold. Substitutions present below 1% mean prevalence were correctly identified as conserved positions with only a few outliers. In the same analysis using a 20% prevalence threshold, substitutions present above 20% mean prevalence were detected in nearly all replicates with less than 0.01% outliers (Appendix A).

### 3.4. Reproducibility

Reproducibility (inter-run variability) was assessed using five replicates of 12 samples and displayed similar results as the repeatability metric. With a single exception, all replicates were successfully sequenced at NS3, NS5a and NS5b (59/60, 98.3%); in one instance, NS5b was not sequenced in one replicate while the remaining genes were analyzed successfully. The median nucleotide and amino acid sequence concordance relative to the consensus sequence was ≥99.86% and ≥99.90%, respectively, for NS3, NS5a and NS5b.

Nearly 100 of all amino acid substitutions with mean prevalence >5% were reproducibly identified at the 2% detection threshold (Figure 3B). Reproducibility was lower for substitutions with a mean prevalence between 2 and 5%, 60% in NS3 and NS5A, and 40% in NS5B. The detection of substitutions with mean prevalence at or below 2% was low as expected, given these residues likely have true frequencies below 2%. Conserved amino acids (<1% mean prevalence) were also identified correctly, with few exceptions. As with the repeatability analysis, we repeated this analysis using a 20% prevalence cut-off for mixtures (Appendix A). Substitutions present above 20% prevalence were detected in most inter-assay replicates but with a greater number of outliers compared to intra-assay replicates. Although a 2% cutoff enables the detection of low prevalence substitutions or minority variants (i.e., <20% prevalence), stochastic effects introduced through RNA extraction, the sampling of small volumes or in RT-PCR efficiency result in the reduced reproducibility of detecting amino acid substitutions when present below 5% prevalence.

### 3.5. Sensitivity–Viral Load

To assess the ability of the genotype-independent assay to reliably amplify and sequence the HCV virus of different genotypes, we evaluated the PCR and sequencing success rate 146 samples consisting of GT 1 to 6 samples. No GT2 samples were available at the time of assessing this validation parameter.

Nearly all GT1 samples with a viral load >5.5 log_10_ IU/mL were successfully sequenced (*n* = 85/86, 98.8%; Table 2). A sequencing success rate of 20–50% was obtained for samples with a viral load <5.5 log_10_ IU/mL. However, only a total of seven samples were tested, four of which have a viral load in the 3-4 log range. Similarly, GT3-6 samples with a viral load >5.5 log_10_ IU/mL were all successfully sequenced (17/17, 100%). In the 23 GT3-6 samples with a viral load between 5.1 and 5.5 log_10_ IU/mL, NS3 and NS5A sequences were obtained in 20 (87%) of the samples, whereas NS5b sequence was generated in only 15 (65%) samples. The drop in sequencing success rates in NS5B is likely due to decreased coverage within NS5b beyond the WG amplicon, primarily in proximity to the amplification primer site. Only half of GT3-6 samples with viral load <5.0 log_10_ IU/mL could be amplified and sequenced. Based on these results, the lower limit of detection was determined to be at >5.0 log_10_ IU/mL.

### 3.6. Sensitivity-Minority Variants

Given the increased sensitivity of NGS methods in detecting minority variants and the high degree of genetic variation amongst HCV samples, we also assessed the assay’s ability to reliably detect these species. This was done by sequencing 95 clinical specimens of multiple genotypes, each processed in duplicate. Upon successful sequencing, the prevalence of each individual nucleotide (A, T, C, G) at every position in the genome was determined for all sequences. Subsequently, for each pair of replicates, the mean and coefficient of variation (CV) of nucleotide prevalence was calculated for each base/position combination. The median and interquartile range of the coefficient of variation was visualized for all observed mean nucleotide prevalence values (Figure 4).

Of the 95 samples selected for analysis in this parameter, 84 were successfully sequenced (88.4%). MiCall genotype assignment using sequence data alone was used for genotype assignment, as no prior LiPA-based genotype result was available for these samples. Sequences were assigned the following genotypes: GT1a (*n* = 69), GT1b (*n* = 10), GT1e (*n* = 3), GT2b (*n* = 2). Note that the 11 specimens that failed sequencing were determined to harbor GT1a HCV after analysis of the partial sequences assembled by the MiCall pipeline.

Through inspection of the relationship between CV and mean nucleotide frequency, we determined that nucleotide substitutions >0.2% prevalence could be reproducibly detected. Minimal variation CV was observed for nucleotide substitutions present in >0.2% of sequence reads, indicating that substitutions observed at these levels in one replicate are likely to be detected in the other replicate at a similar prevalence. However, nucleotide detection becomes less reliable for rare substitutions with <0.2% prevalence as shown by the sharp increase in CV at lower prevalence levels. However, a more conservative threshold may be warranted given MiSeq error rates may be higher. In a separate analysis of two HCV clones combined at different ratios (data not shown), we estimated the combined MiSeq random and systematic error rate to be approximately 0.5%. Given the greater error rate, as well as the limitations of PCR, the technical lower limit of minority species detection should be considered 0.5%.

### 3.7. Specificity

Across all analyses included in this validation, there was no evidence of contamination observed. Neither the HCV-negative sample nor the negative controls (DEPC-treated water) included in runs of this validation, as well routine clinical runs following this study, produced a detectable PCR product by agarose gel electrophoresis. Common interfering substances (e.g., hemoglobin) are unlikely to affect our assay due to the RNA extraction method.

Analysis for interfering substances was not performed during this study but was previously done during the validation of the GT1-optimized method. In that study, the lack of interference from other common viruses was assessed using HCV-positive plasma was spiked with one of three additional specimens: a clinically derived HIV-1 plasma sample, a HIV-1 molecular clone or a clinically-derived HBV-positive plasma sample. These mixed-infection samples were subsequently processed in triplicate using the GT1-optimized method. No evidence of the off-target amplification of HIV or hepatitis B virus from spiked samples was observed in that study, nor in coinfected samples using the newly described method since its clinical implementation [36].

### 3.8. Genotype Spectrum

A total of 146 samples, representing 7 different genotypes and subtypes, were used to evaluate the ability of this assay to classify genotypes against the previously reported genotype. Roughly 90% sequencing success rates were observed for the epidemic strains present in G7 countries, such as GT1a, GT1b, 2b, 3a, and GT6 (Table 3). About 70% success rates were observed in GT4 and GT5. Two of the 55 GT3-6 samples were excluded in the genotype spectrum analysis due to viral loads <5 log_10_ IU/mL. One other sample from the presumed GT3-6 sample set, which had no available LiPA-based genotype result, was identified as GT1b using sequence data, leaving 52 GT3-6 specimens. In total, NS3 and NS5a were both successfully sequenced in 45 out of 52 GT3-6 samples (87%), and NS5b was successfully sequenced in 39/52 (75%) samples in these genotypes. Small drops in coverage at this position are to be expected, given its proximity to the location of the 2nd round PCR amplification primer position (NS5b codon 336) and the decreased efficiency of Nextera XT processing near the ends of the amplicon. All samples failing NS5b sequencing, but not NS3 and NS5a, had viral loads >5.1 log_10_ IU/mL.

We next evaluated the ability of the sequencing assay to accurately identify the HCV genotypes of the 146 samples described above. Genotypes inferred from the sequence data by MiCall were compared with the LiPA-based genotype data provided by the sample providers where available. Genotype comparisons could be assessed for 95 samples, as our study was blinded to the LiPA-based genotypes for 37 samples originating from the C-WORTHY trial. There was a 100% concordance between the genotype calls provided by the LiPA probe-based assay and the genotype calls determined by direct MiSeq sequence mapping. Subtype information was identical for all genotype 1–3 samples across both genotyping methods. In cases of genotypes 4–6 (*n* = 18), ambiguous subtype information (e.g., genotype 4 or 4a/c/d) was provided by LiPA for all genotype 4 and 6 samples (*n* = 13/18). Direct sequence mapping provided definitive subtype information (e.g., genotype 4a) for all but three of these samples (*n* = 10/18), where the remaining three could not be sequenced.

### 3.9. Overlapping Region of Amplicons

The MiDi amplicon was designed to provide sequence coverage to the end of NS5b, beyond the WG amplicon that extends from Core to codon 336 of NS5b. However, only two RAS positions are covered by the overlap of WG and MiDi amplicons, and no sequences with known resistance-associated mutations at these positions were observed in this validation (Appendix A). We therefore assessed the proportion of each nucleotide at all overlapping positions in NS5b (not only RAS positions). There was an excellent correlation between the proportion each nucleotide observed between MiDi and WG (r^2^ > 0.98). A summary of the mean number of sequence reads, amino acid counts and amino acid prevalence for the two NS5B RAS (S282 and L320) within the overlapping region obtained by WG and MiDi amplicons is shown in Appendix A.

## 4. Discussion

We had previously developed HCV genotyping assays that were optimized to detect the NS3 Q80K polymorphism in HCV GT1 specimens only, using both Sanger and NGS (MiSeq) sequencing technologies [36]. Although these assays are suitable to capture the resistance profile of the most prevalent HCV genotype, GT1 represented only 44% of all new HCV infections globally in 2015 [40]. Despite the availability of HCV RAS screening laboratory services in the United States (e.g., Labcorp, Quest Diagnostics), these have only been validated for GT1 infections; no commercially available test to our knowledge has been extensively validated to characterize non-GT1 HCV infections. We have therefore developed and validated an NGS method to infer HCV genotypes and identify RAS for clinical purposes regardless of the associated HCV genotype/subtype classification. This genotype-independent method can be used to screen for all currently known RAS within NS3, NS5a and NS5b and potentially guide therapeutic decision making in treatment-naïve and -experienced individuals.

The genotype-independent sequencing assay described here demonstrated similar performance as the previously validated GT1-optimized NGS method [36]. Overall, high accuracy and precision in all replicates tested was observed (>99.7%). Importantly, samples used to assess both inter- and intra-assay concordance were highly diverse as illustrated by the number of low-prevalence (2–5%) nucleotide mixtures observed in sequences from these samples. Despite this intra-sample diversity, NGS sequencing was capable of accurately and reproducibly characterizing the diversity of HCV species within a sample, regardless of genotype. Note that reproducibility displayed similar results as the repeatability metric, albeit with slightly more variability at both the 2% and 20% prevalence thresholds for amino acid substitution detection.

Substitution prevalence thresholds used in this study focused on two cutoffs: 20% to reflect a Sanger-sequencing limit of detection and 2% to utilize the potential of NGS to detect minority variants. A suitable question stemming from this study is which cutoff is most optimal for the detection of RAS and is sufficient to correlate with clinical outcomes. A separate study found that a 10% cutoff predicted DAA response in GT3 infections [41]. Parameters in this study intended to analyze the assay’s capacity to detect substitutions consistently and quantitatively. Results show that when a prevalence cutoff is utilized to make a nucleotide or amino acid call, the most variance in detecting substitutions is centered around the cutoff itself, as one would expect. Using a 20% cutoff, effectively all high-prevalence (i.e., ≥20%) substitutions are detected consistently, and no substitutions with a true prevalence below 10% are detected. Substitutions whose prevalence is between 10 and 20% are only detected incorrectly after repeat testing of the same sample, on average in 1 of 5 replicates. In contrast, utilizing a 2% cutoff shows a greater spread in its ability to detect substitutions around and above the threshold. Generally, only substitutions with a presumed prevalence above 10% could be reliably detected despite the lower cutoff. However, further studies are needed to elucidate on potential clinical implications of modifying these prevalence cutoffs and the effect of minority variants on treatment outcomes.

In this study, sample providers provided LiPA-based genotype classifications for samples, where available. In rare cases, the LiPA assay can misclassify infections or fail to resolve genotype and/or subtype entirely [42,43,44,45]. By using a set of pan-genomic PCR primers and inferring HCV genotype in silico from deep-sequencing genomic data, this assay obviates the need to perform HCV genotyping by LiPA prior to sequencing and potentially resolves HCV subtypes with greater accuracy. Laboratories may also opt to perform NS5b sequencing for genotype identification rather than LiPA assays or whole-genome sequencing assays such as the one described here. NS5B sequencing has demonstrated lower rates of genotype misclassification compared to LiPA and other genotyping kits, and may also be more cost-effective [18]. However, this approach restricts genomic data to the NS5b region only and fails to capture RAS located outside this gene.

This whole-genome sequencing assay can be used to better characterize the prevalence of RAS across the HCV genome and within non-GT1 HCV infections. Data generated by this method, in conjunction with phenotype and observational studies, can be used to identify novel RAS within or outside of these genes and possibly correlate with treatment outcomes. Additional validation studies would be required to reassess the performance of this assay for those new mutations.

Having demonstrated similar performance to a previously validated GT1 assay, this genotype-independent HCV sequencing assay further expands the availability of RAS screening to other HCV genotypes. This method has been implemented clinically at the BC Centre for Excellence in HIV/AIDS and has been used to deliver ~4000 resistance test reports to physicians across Canada as of August 2021. The assay protocols and paired software are freely available to interested labs worldwide; software is available at https://github.com/cfe-lab/MiCall.

Both the validated GT1-optimized method and new genotype-independent method exhibited similar limitations. In terms of limit of detection, both assays require significantly more input starting material (~5log_10_ IU/mL) than their Sanger sequencing counterparts [36]. Despite this, the plasma viral load required for testing is below the expected HCV viral load range for most untreated individuals, including those living with HIV, cirrhosis, advanced liver disease, or those who have previously failed DAA treatment [46,47,48,49,50]. Another limitation of the assay described here is the requirement for two amplicons to cover the entire HCV genome and capture the full spectrum of HCV genotype, except for the 5′ and 3′ untranslated regions. The generation of the additional MiDi amplicon to cover the remainder of NS5B beyond codon 336 has increased cost and laboratory work implications. However, this additional cost may be warranted due to the potential for RAS beyond codon 336 and the use of NS5b sequence data for phylogenetic cluster identification [51].

A notable consideration of this study is the genotype representativeness within individual validation parameters. With the exception of sensitivity and genotype coverage that were analyzed using a large spectrum of available genotypes, the remaining validation tests were assessed using predominantly GT1 specimens. Additionally, these consisted largely of GT1a samples, with the minimal inclusion of GT1b specimens. Equal representativeness of all genotypes in each validation parameter was not possible in this validation and represents an important limitation. GT1b and non-GT1 samples, particularly rare genotypes, could not be included in the study in large numbers due to a lack of sample availability. These factors may limit our ability to infer assay performance from this study for these uncommon genotypes. Although many genotypes are under-represented in this study, the assay demonstrated acceptable metrics in analyzing the available specimens. Since utilizing this method clinically at the BC Centre for Excellence in HIV/AIDS, no negative bias has been observed towards these less common genotypes (data not shown).

## 5. Conclusions

In summary, we have developed and characterized a genotype-independent, whole-genome HCV sequencing workflow on the Illumina MiSeq platform. This assay can provide accurate HCV drug resistance profiles to currently known RAS, notably in cases of DAA-treatment failure or in select treatment-naïve cases, to personalize HCV therapy. We demonstrated positive and consistent performance of the test including accuracy, repeatability, reproducibility and sensitivity. Importantly, the assay successfully amplified and sequenced 90% of samples regardless of genotype. Since the method generates near-full-genome coverage for any HCV genotype, the method could be expanded to screen for other known RAS and phylogenetic cluster identification. The assay and associated bioinformatic software also simultaneously provide genotype classification, which can supplement validated, probe-based genotyping results or be used on a research-use-only basis in instances wherein no genotype assignments are otherwise available. Together, this assay and bioinformatic pipeline can also facilitate the implementation of HCV drug resistance testing by other laboratories globally, with an end-to-end, sample-to-report workflow.

## Figures and Tables

**Figure 1 viruses-13-01721-f001:**
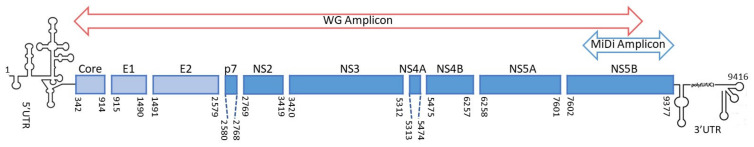
HCV genome map. A schematic representation of the HCV genome depicting coverage of the genotype-independent, near-whole genome sequencing method wherein the genome is amplified in two fragments then sequenced in parallel.

**Figure 2 viruses-13-01721-f002:**
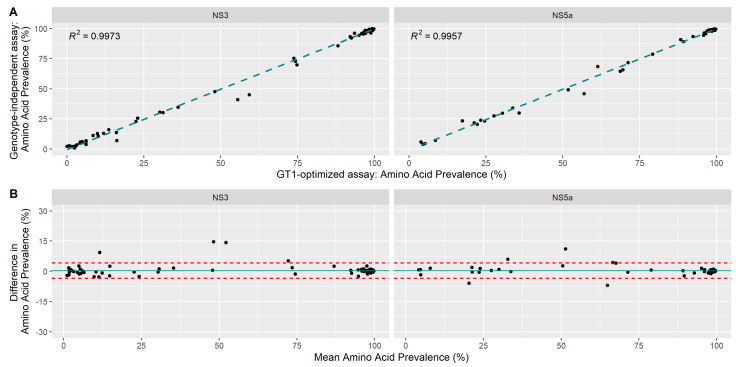
Accuracy of RAS quantification using the genotype-independent sequencing method compared to a genotype 1-optimized assay. Accuracy for NS3 and NS5a was assessed by the comparison of amino acid substitution frequencies at predefined, resistance-associated position as determined by our previously validated genotype 1-optimized and genotype-independent assays. Each point represents one possible RAS; note that positions with multiple possible amino acid residues are represented as separate data points. (**A**) Amino acid prevalence is compared between both sequencing methods for the same samples. (**B**) The difference in RAS frequencies between both sequencing methods as a function of mean RAS prevalence. The two red dotted lines represent the 95% limits of agreement.

**Figure 3 viruses-13-01721-f003:**
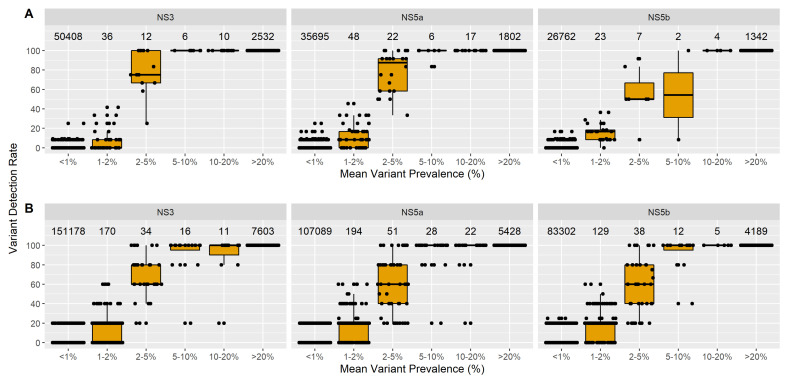
Repeatability and reproducibility of amino acid substitutions in NS3, NS5a and NS5b. Substitutions observed at a prevalence ≥2% in a given replicate were considered “detected”. Variant detection rate (defined as the % of replicates per sample, in which a substitution was detected) is categorized by the mean prevalence of a substitution across all replicates of a sample. Replicate testing of each sample began from the same RNA extract; all steps beginning from the RT-PCR were repeated. Numbers indicated on the top of each graph represent the total number of substitutions in each bin. (**A**) Repeatability of amino acid substitutions was determined using four samples (3 GT1a, 1 GT1b) in 12 replicates processed on a single MiSeq run. (**B**) The reproducibility of amino acid substitutions was determined using 12 samples (10 GT1a, 1 GT1b, 1 GT3) in 5 replicates processed on separate days on five separate MiSeq runs.

**Figure 4 viruses-13-01721-f004:**
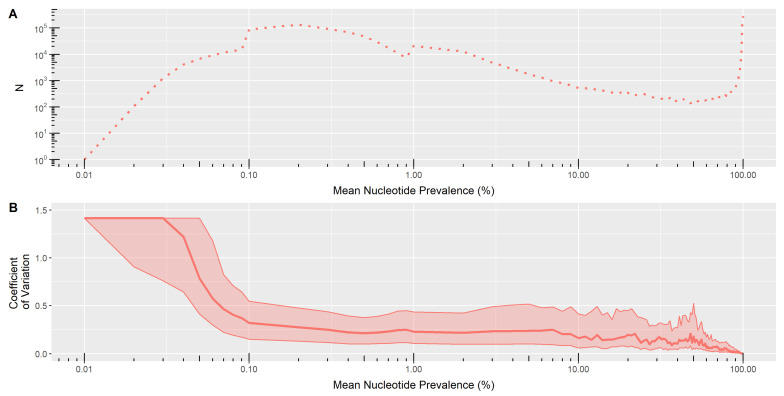
Assay sensitivity in detecting minority variants between sample duplicates at different mean nucleotide frequencies (defined as the mean prevalence of all permutations of nucleotides and positions within a sequence). This parameter was assessed using 95 samples run in duplicate, of which 84 passed sequencing. (**A**) The total number of nucleotide substitutions (N) detected at a particular mean substitution prevalence. (**B**) Median coefficient of variation (CV) of minor species with interquartile range (IQR) at different mean substitution prevalence.

**Table 1 viruses-13-01721-t001:** Specimen characteristics for assessing individual assay validation parameters.

Test	No. of Samples	Replicates per Sample	HCV pVL log_10_ (IU/mL), Median (Range)	Genotypes (LiPA Assay)
Accuracy	93	2	6.8 (3.0–7.6)	1a (*n* = 78) 1b (*n* = 10) 1e (*n* = 3) 2b (*n* = 2)
Repeatability	4	12	6.5 (6.1–7.3)	1a (*n* = 3) 1b (*n* = 1)
Reproducibility	12	5	6.5 (6.0–7.7)	1a (*n* = 10) 1b (*n* = 1) 3 (*n* = 1)
Sensitivity (viral load) & Genotype coverage	148	1	6.1 (3.0–7.6)	1a (*n* = 78) 1b (*n* = 10) 1e (*n* = 3) 2b (*n* = 2) 4 (*n* = 8) 5 (*n* = 5) 6 (*n* = 5) Unknown (*n* = 37)
Sensitivity (minor species)	95	2	NA	Unknown (*n* = 95)
Specificity-Negatives	5	3	NA	NA

HCV, Hepatitis C Virus; pVL, plasma viral load; IU, international units; LiPA, line probe assay; NA, not available. Dashes indicate no data were available for the associated parameter.

**Table 2 viruses-13-01721-t002:** Sequencing success of the genotype-independent NGS assay in GT1 and GT3-6 sample sets stratified by viral load category.

	Genotype 1 (*n* = 93)			Genotypes 3–6 (*n* = 53) ^^^		
HCV pVL log_10_ (IU/mL)	NS3	NS5A	NS5B	NS3	NS5A	NS5B
>7	23/23 (100%)	23/23 (100%)	23/23 (100%)	2/2 (100%)	2/2 (100%)	2/2 (100%)
6.6–7.0	32/33 (97%)	32/33 (97%)	32/33 (97%)	1/1 (100%)	1/1 (100%)	1/1 (100%)
6.1–6.5	14/14 (100%)	14/14 (100%)	14/14 (100%)	4/4 (100%)	4/4 (100%)	4/4 (100%)
5.6–6.0	16/16 (100%)	16/16 (100%)	16/16 (100%)	10/10 (100%)	10/10 (100%)	10/10 (100%)
5.1–5.5	1/2 (50%)	1/2 (50%)	1/2 (50%)	20/23 (87%)	20/23 (87%)	15/23 (65%)
<5	1/5 (20%)	2/5 (40%)	1/5 (20%)	7/13 (7/54%)	7/13 (54%)	7/13 (54%)

HCV, hepatitis C virus; pVL, plasma viral load; IU, international units. Genotypes were inferred through sequence analysis or in the absence of sequence data from assigned genotypes from LiPA assays. HCV subtype information was obtained using phylogenetic analyses based on MiSeq data. Sequencing success was defined as sequences having a minimum of 100-fold coverage at all RAS-associated positions in each respective gene. ^ Two genotype 3–6 samples from the set of 55 samples were excluded from this table, as no viral load data were available.

**Table 3 viruses-13-01721-t003:** Sequencing success for the HCV NS3, NS5a and NS5b genes using the genotype-independent-HCV assay for genotypes 1–6.

Genotype	Genotype Subtypes *	Samples Attempted ^^^	Sequenced (%)		
			NS3	NS5a	NS5b
1	1a	78	72 (92%)	72 (92%)	72 (92%)
	1b	11	11 (100%)	11 (100%)	11 (100%)
	1e	3	2 (67%)	2 (67%)	2 (67%)
2	2b	2	2 (100%)	2 (100%)	2 (100%)
3	3a	20	17 (100%)	17 (100%)	17 (100%)
4	4a, 4n	7	5 (71%)	5 (71%)	2 (29%)
5	5a	4	3 (75%)	3 (75%)	3 (75%)
6	6a, 6e, 6h, 6k, 6t	21	20 (95%)	20 (95%)	17 (85%)
Overall	-	146	132 (90%)	132 (90%)	126 (86%)

* Genotypes were inferred through sequence analysis or in the absence of sequence data from assigned genotypes from LiPA assays. HCV subtype information was obtained using phylogenetic analyses based on MiSeq data. Sequencing success was defined as sequences having a minimum of 100-fold coverage at all RAS-associated positions in each respective gene. ^^^ Only samples with viral loads above the estimated assay limit of detection (5 log_10_ IU/mL) were included.

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
