# Peer review of "Validation of a Genotype-Independent Hepatitis C Virus Near-Whole Genome Sequencing Assay"

_viruses, 2021, doi:10.3390/v13091721_

Round 1

Reviewer 1 Report

The article is interesting as it allows the development of a trial that helps the clinician in making decisions in patients undergoing treatment with HCV.

Some minor issues should be reviewed by the authors.

-Reduce and explain acronyms in the abstract.

-Tables should be self-explanatory with their table footer.

-The objective of the study is out of focus.

-The sample size of the GT1b genotype analyzed is not very high, this should be an indication of study limitations together with the rest of the rare genotypes.

-Review typographical errors.

-An outline of the global work carried out would be advisable for a more agile reading of the manuscript.

Reviewer 2 Report

In this manuscript Lapointe et al describe a sequence protocol to genotype HCV DAA resistant mutations present in infected individuals. The method uses next generation massive sequencing to sequence almost full-length HCV genomes of genotype 1-6. The authors claim literally that this assay can help clinicians and patients to make treatment decisions. First, the approach is not novel since there are several, if not many, described high-throughput assays to genotype HCV DAA resistance, it is unclear what is novel here. Second, the authors sequenced samples from HCV infected individuals failing DAA therapy (line 81); however, they fail to show whether individual virus sequences correlate with therapy failure, otherwise, sequencing will be useless to take clinical decisions. Moreover, the authors do not show any sequence. Have the authors deposited the study sequences in any database repository? As stated by the authors, there are now pan-genotypic alternatives to treat infected individuals (line 52), with these pan-genotypic options it is unclear the utility of HCV genome sequencing.

Reviewer 3 Report

The manuscript must undergo some changes in order to reconsider its acceptance. The proposed changes or suggestions are as follows:

- In the material and methods section, a brief description of the study population as well as the dates of performance are missing.

- Almost no samples of genotype other than GT-1 are included in the accuracy and precision tests of the technique. Since we are evaluating a genotype-independent assay, it may be of interest that the proportion of genotypes is more equal.

- The sensitivity assays of the technique include 55 samples with genotypes 3-6 and in 37 no genotype is recognized. This does not show a high sensitivity.

- It should specify why LiPA is considered as a reference technique for genotyping when the assay being evaluated is based on sequencing.

- Regarding the specificity assay, 95 samples are used and no genotype is detected in any case?

- At page 7, lines 294-295: The error of lack of reference source should be corrected.

-At page 8, lines 340-341: The error of lack of reference source should be corrected.

- At page 9, line 361: The error of lack of reference source should be corrected.

- At page 9, line 367: The error of lack of reference source should be corrected.

- At page 12, line 452: It refers that 2 of the 55 samples with GT 3-6 were excluded due to low viral load and then refers that NS3 and NS5a were successfully sequenced in 45 of the 52 samples and NS5b in 39/52; what happened to the missing sample? This point in the results needs to be better clarified.

- Line 563: Perhaps it would be an important improvement in the study to expand data on underrepresented genotypes.

Round 2

Reviewer 2 Report

Unfortunately, the authors did not address my main concerns to this manuscrip. First, it is unacceptable not to publically show the data in which this study is based, that is, the study sequences. Second, there is not virology in this study, the utility of the method shown here is suggested but not demonstrated, I would suggest the authors to try a methods journal rather than a virology journal.

Reviewer 3 Report

The authors provide changes to all requests and suggestions. Despite the limitations of the study, I consider the paper is now suitable for publication.